# Association of Results of Four Lateral Flow Antibody Tests with Subsequent SARS-CoV-2 Infection

Lucy Findlater,[a,b] Adam Trickey,[c] Hayley E. Jones,[b,c] Amy Trindall,[a] Sian Taylor-Phillips,[d] Ranya Mulchandani,[a] EDSAB-HOME Investigators, Isabel Oliver,[a,b] David Wyllie[a]

[a]UK Health Security Agency, Cambridge, United Kingdom
[b]National Institute of Health Research Health Protection Research Unit on Behavioural Science and Evaluation at the University of Bristol, Bristol, United Kingdom
[c]Population Health Sciences, University of Bristol, Bristol, United Kingdom
[d]University of Warwick, Coventry, United Kingdom

**ABSTRACT** Severe acute respiratory syndrome coronavirus 2 (SARS-CoV-2) vaccine coverage remains incomplete, being only 15% in low-income countries. Rapid point-of-care tests predicting SARS-CoV-2 infection susceptibility in the unvaccinated may assist in risk management and vaccine prioritization. We conducted a prospective cohort study in 2,826 participants working in hospitals and Fire and Police services in England, UK, during the pandemic (ISRCTN5660922). Plasma taken at recruitment in June 2020 was tested using four lateral flow immunoassay (LFIA) devices and two laboratory immunoassays detecting antibodies against SARS-CoV-2 (UK Rapid Test Consortium's AbC-19 rapid test, OrientGene COVID IgG/IgM rapid test cassette, SureScreen COVID-19 rapid test cassette, and Biomerica COVID-19 IgG/IgM rapid test; Roche N and Euroimmun S laboratory assays). We monitored participants for microbiologically confirmed SARS-CoV-2 infection for 200 days. We estimated associations between test results at baseline and subsequent infection, using Poisson regression models adjusted for baseline demographic risk factors for SARS-CoV-2 exposure. Positive IgG results on each of the four LFIAs were associated with lower rates of subsequent infection with adjusted incidence rate ratios (aIRRs) of 0.00 (95% confidence interval, 0.00 to 0.01), 0.03 (0.02 to 0.05), 0.07 (0.05 to 0.10), and 0.09 (0.07 to 0.12), respectively. The protective association was strongest for AbC-19 and SureScreen. The aIRR for the laboratory Roche N antibody assay at the manufacturer-recommended threshold was similar to those of the two best performing LFIAs at 0.03 (0.01 to 0.10). Lateral flow devices measuring SARS-CoV-2 IgG predicted disease risk in unvaccinated individuals over a 200-day follow-up. The association of some LFIAs with subsequent infection was similar to laboratory immunoassays.

**IMPORTANCE** Previous research has demonstrated an association between the detection of antibodies to SARS-CoV-2 following natural infection and protection from subsequent symptomatic SARS-CoV-2 infection. Lateral flow immunoassays (LFIAs) detecting anti-SARS-CoV-2 IgG are a cheap, readily deployed technology that has been used on a large scale in population screening programs, yet no studies have investigated whether LFIA results are associated with subsequent SARS-CoV-2 infection. In a prospective cohort study of 2,826 United Kingdom key workers, we found positivity in lateral flow test results had a strong negative association with subsequent SARS-CoV-2 infection within 200 days in an unvaccinated population. Positivity on more-specific but less-sensitive tests was associated with a markedly decreased rate of disease; protection associated with testing positive using more sensitive devices detecting lower levels of anti-SARS-CoV-2 IgG was more modest. Lateral flow tests with high specificity may have a role in estimation of SARS-CoV-2 disease risk in unvaccinated populations.

**KEYWORDS** cohort study, immunity, lateral flow device, SARS-CoV-2

Address correspondence to David Wyllie, david.wyllie@phe.gov.uk.
The authors declare no conflict of interest.

10.1128/spectrum.02468-22 **1**

The COVID-19 pandemic has caused a global health crisis. Worldwide, as of 14 April 2022, there have been over 490 million cases and 6 million deaths associated with severe acute respiratory syndrome coronavirus 2 (SARS-CoV-2) (1). The infection is common worldwide, and effective vaccines have now been developed and distributed, most widely in high-income countries (2). However, while about 65% of the world's population has now received at least one vaccine dose, in low-income countries, the corresponding figure is 20% (3). Additionally, vaccine hesitancy is widespread globally (4, 5). Understanding individual risk of infection impacts hesitancy and other aspects of individual behavior and is helpful for population surveillance and pandemic planning and response. In vaccinated populations, understanding individual risk could also help to monitor immune responses to vaccination and inform prioritization of booster delivery (5–8).

Antibodies to the SARS-CoV-2 spike and nucleocapsid proteins are generated in over 90% of individuals with symptomatic infection and persist for months (9–14). Multiple studies following up individuals with SARS-CoV-2 infection have described protection from SARS-CoV-2 following natural infection in individuals with detectable antibody levels: protection from subsequent symptomatic infection with SARS-CoV-2 was estimated at about 85% protection in two overlapping meta-analyses of 19 studies performed in the general population, health care workers, college students, and residents in long-term care facilities (15, 16). In some studies, quantitative antibody levels were recorded, and increased protection in individuals with higher antibody levels was observed (10, 17).

A range of laboratory-based immunoassays and lateral flow immunoassays (LFIAs) have been developed to detect anti-SARS-CoV-2 IgG or IgM antibodies (18–20). LFIAs are small devices that allow antibody testing without the need for a laboratory (19, 21–23) and have been deployed at population scale (24). Previous work has explored the sensitivity and specificity of LFIAs to detect antibody responses to the spike or nucleoprotein antigens of SARS-CoV-2 (20–22) and has shown that LFIA sensitivities vary (21, 25). Importantly, in a large-scale comparative study of the accuracy of four LFIAs, we observed a trade-off between sensitivity and specificity, with two devices being more sensitive and two more specific (21). All four devices studied in this work were more likely to give positive results for samples with high, rather than low, levels of antibodies following natural infection, and more specific devices detected low levels of antibodies less often than less specific devices (21).

The potential of lateral flow devices to predict individual risk of SARS-CoV-2 infection has not been evaluated. In this study, our objective was to quantify directly the association of the results of four lateral flow antibody tests (Rapid Test Consortium's AbC-19 rapid test, OrientGene COVID IgG/IgM rapid test cassette, SureScreen COVID-19 rapid test cassette, and Biomerica COVID-19 IgG/IgM rapid test) and two laboratory-based immunoassays (Roche Elecsys and Euroimmun Anti-SARS-CoV-2 enzyme-linked immunosorbent assay [ELISA]), with subsequent symptom-driven PCR test positivity. We did this in a cohort of 2,826 unvaccinated United Kingdom keyworkers recruited to the Evaluating Detection of SARS-CoV-2 AntiBodies at HOME (EDSAB-HOME) study (21, 22, 26, 27) and who were followed up for 200 days.

## RESULTS

**SARS-CoV-2 positivity.** The study follow-up period ran until 24 January 2021. After a period with low levels of infection (period 1), two waves of intense SARS-CoV-2 transmission in England occurred (periods 2 and 3) (Fig. 1) (28). During follow-up, 285/2,826 (10%) of participants had positive SARS-CoV-2 reverse transcriptase PCR (RT-PCR) tests (Table 1). Symptom data were available on 216, of whom 15 (7%) reported having no symptoms in questionnaires when they were tested.

Overall, 3/285 (1%) PCR positives were reported in period 1, 109/285 (38%) in period 2, and 173/285 (61%) in period 3. Kaplan-Meier survival curves for each LFIA and laboratory assays are shown in Fig. 2. The crude rate of PCR positivity among antibody-positive participants was lowest for AbC-19 (0 observed events, estimated rate with Firth penalization of 0.1 per 100 person-years [95% confidence interval (CI), 0.0 to 2.2]) followed by Roche (0.6

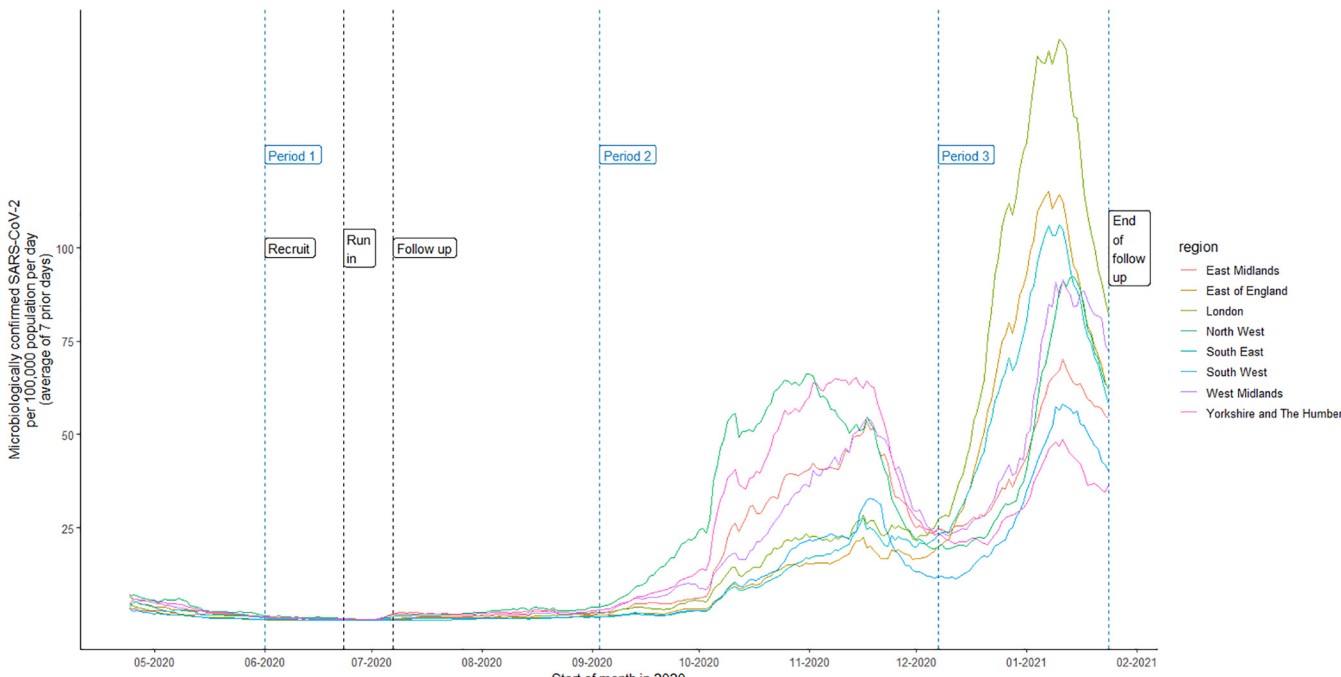

**FIG 1** SARS-CoV-2 infections in England during the study period. National data showing the 7-day moving average of the case rate per 100,000 people, stratified by region, throughout the study follow-up time. We split this into three periods based on the different waves of infection in the population as follows: period 1, 1 June 2020 to 2 September 2020; period 2, 3 September 2020 to 6 December 2020; period 3, 7 December 2020 to 24 January 2021. Recruitment took place from 1 June 2020 to 30 June 2020. There was a run-in period of 2 weeks after recruitment where any positive SARS-CoV-2 results were excluded to ignore individuals who were already infected. Follow-up was for 200 days post recruitment. Data on population size were obtained from the Office for National Statistics. Data on SARS-CoV-2 cases were obtained from the UK Government.

[95% CI, 0.2 to 2.2]), SureScreen (0.7 [95% CI, 0.2 to 2.4]), Euroimmun (0.9 [95% CI, 0.3 to 2.6]), OrientGene (1.6 [95% CI, 0.7 to 3.4]), and Biomerica (2.3 [95% CI, 1.2 to 4.3]).

**Association of results and subsequent SARS-CoV-2 infection.** The adjusted rate ratio for PCR positivity was lowest for the AbC-19 device (0.00 [95% CI, 0.00 to 0.01]) followed by Roche (0.02 [95% CI, 0.01 to 0.04]), SureScreen (0.03 [95% CI, 0.02 to 0.05]), Euroimmun (0.04 [95% CI, 0.03 to 0.06]), OrientGene (0.07 [95% CI, 0.05 to 0.10]), and Biomerica (0.09 [95% CI, 0.07 to 0.12]) but with overlapping CIs (Table 1; Fig. 3).

**Antibody positivity thresholds.** Since the sensitivity of lateral flow devices for the detection of low levels of SARS-CoV-2 antibodies is known to differ between devices, we explored how the association varied when alternative test positivity thresholds were used for the laboratory immunoassays (21). In Fig. 4, we present the adjusted incidence rate ratios for both Roche and Euroimmun laboratory-based immunoassays, dichotomized at various different assay signals. The strength of association between test positivity and subsequent infection was somewhat stronger when higher positivity thresholds were used (Fig. 4).

## DISCUSSION

This study demonstrates that participants who tested positive for SARS-CoV-2 antibodies with any of the four lateral flow immunoassays (AbC-19, SureScreen, OrientGene, and Biomerica) or two laboratory immunoassays (Roche and Euroimmun) detecting anti-SARS-CoV-2 antibodies had a lower rate of subsequent SARS-CoV-2 infection than individuals who did not have detectable antibodies. Adjusted incidence rate ratio (95% CI) estimates for disease were 0.00 (0.00, 0.01) and 0.03 (0.02, 0.05) for the two most predictive lateral flow devices.

While this study shows a strong association between lateral flow device results and subsequent SARS-CoV-2 infection, it has several limitations. Firstly, it applies to a historical cohort of unvaccinated individuals and prior to the emergence of variants such as Delta, Omicron, and BA.5 (29). The performance of LFIAs might be different in the context of

**TABLE 1** Numbers of SARS-CoV-2 PCR tests positive by antibody status[a]

| | Antibody test negatives | | | | | | | Antibody test positives | | | | | | | | |
| PCR test | No. testing antibody negative | No. of PCR positives in period 1 | No. of PCR positives in period 2 | No. of PCR positives in period 3 | Total no. of PCR positives | Person yrs at risk | Observed rate of PCR positivity per 100 person yrs with Firth penalization (95% CI) | No. testing antibody positive | No. of PCR positives in period 1 | No. of PCR positives in period 2 | No. of PCR positives in period 3 | Total no. of PCR positives | Person yrs at risk | Observed rate of PCR positivity per 100 person yrs with Firth penalization (95% CI)[c] | Unadjusted rate ratio with Firth penalization (95% CI) | Adjusted[b] rate ratio with Firth penalization (95% CI) |
|---|---|---|---|---|---|---|---|---|---|---|---|---|---|---|---|---|
| AbC-19 | 2,261 | 3 | 109 | 173 | 285 | 1,456.7 | 19.6 (17.5–22.0) | 565 | 0 | 0 | 0 | 0 | 370.1 | 0.1 ↑ (0.0–2.2) | 0.00 (0.00–0.02) | 0.00 (0.00–0.01) |
| SureScreen | 2,269 | 3 | 109 | 171 | 283 | 1,462.0 | 19.4 (17.3–21.8) | 557 | 0 | 0 | 2 | 2 | 364.7 | 0.7 (0.2–2.4) | 0.03 (0.02–0.05) | 0.03 (0.02–0.05) |
| OrientGene | 2,203 | 3 | 107 | 169 | 279 | 1,419.3 | 19.7 (17.5–22.1) | 623 | 0 | 2 | 4 | 6 | 407.5 | 1.6 (0.7–3.4) | 0.08 (0.06–0.11) | 0.07 (0.05–0.10) |
| Biomerica | 2,186 | 3 | 103 | 170 | 276 | 1,409.0 | 19.6 (17.4–22.1) | 640 | 0 | 6 | 3 | 9 | 417.8 | 2.3 (1.2–4.3) | 0.10 (0.07–0.13) | 0.09 (0.07–0.12) |
| Roche | 2,219 | 3 | 108 | 172 | 283 | 1,429.4 | 19.8 (17.7–22.3) | 607 | 0 | 1 | 1 | 2 | 397.3 | 0.6 (0.2–2.2) | 0.03 (0.02–0.05) | 0.02 (0.01–0.04) |
| Euroimmun | 2,237 | 3 | 108 | 171 | 282 | 1,441.1 | 19.6 (17.4–22.0) | 589 | 0 | 1 | 2 | 3 | 385.6 | 0.9 (0.3–2.6) | 0.05 (0.03–0.07) | 0.04 (0.03–0.06) |

[a]Number of SARS-CoV-2 RT-PCR positives and observed rate of PCR positivity among participants who tested antibody negative and participants who tested antibody positive with each LFIA and laboratory immunoassay (n = 2,826).
[b]Adjusted for sex, ethnicity (white or nonwhite), age (continuous), high-risk occupation (medical or nursing staff), region, average weekly SARS-CoV-2 infection incidence rate in NHS region of residence of participant.
[c]The symbol ↑ indicates the crude observed rate before application of Firth penalization of 0.0.

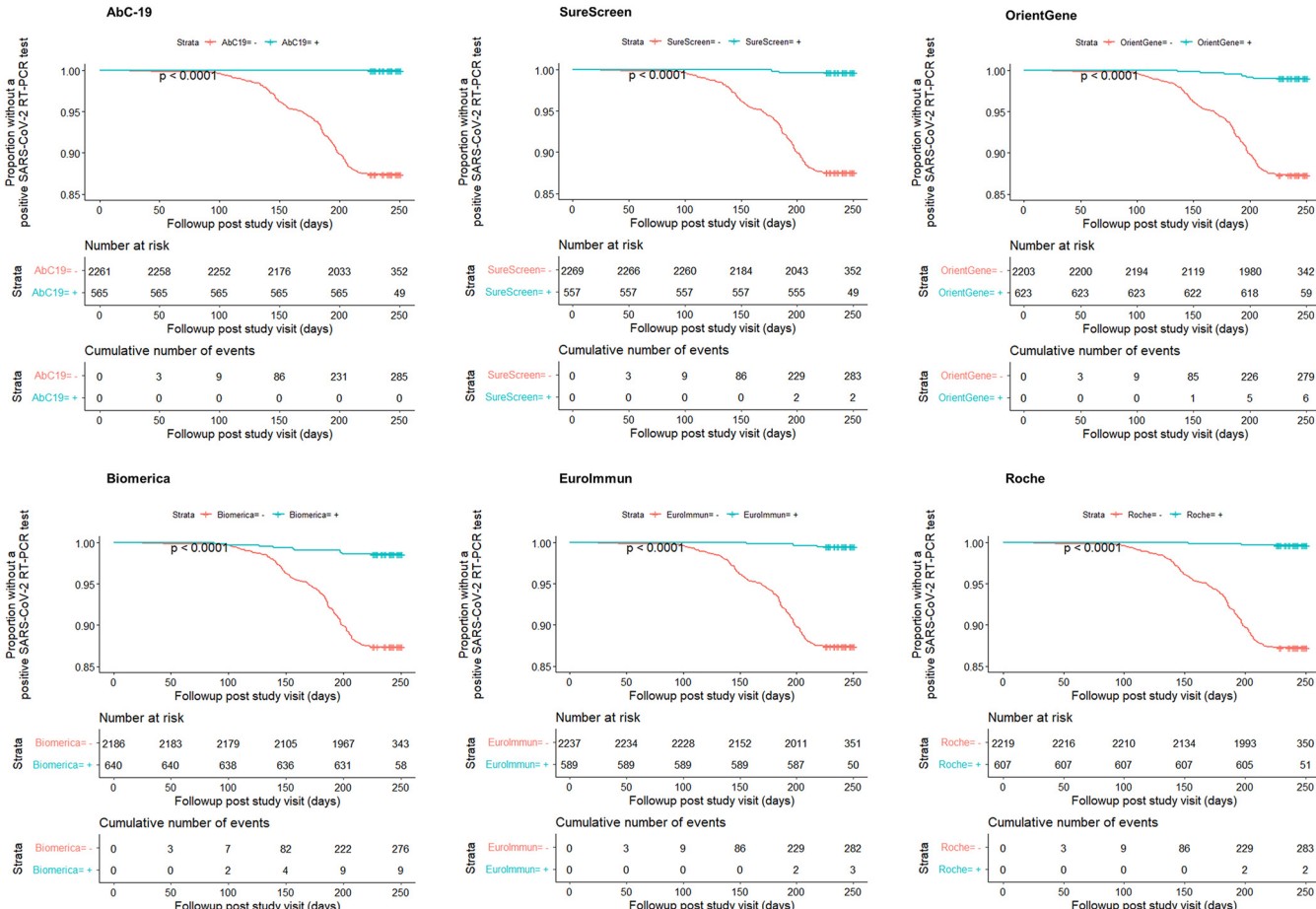

**FIG 2** Time to positive SARS-CoV-2 RT-PCR test stratified by antibody test result for each LFIA. Kaplan-Meier survival curves showing the proportion of participants without a positive SARS-CoV-2 RT-PCR test throughout the follow-up period, stratified by antibody test results for each lateral flow immunoassay.

immunity gained from vaccination (in addition to or instead of natural infection) and to estimate protection against different circulating SARS-CoV-2 variants with differential antibody neutralization; this is an important area for future research (8).

Secondly, lateral flow devices were applied and read using stored plasma, obtained by venesection, in a laboratory setting by trained professionals rather than relying on measurement in the field by the public using finger-prick technology. This arrangement had the advantage that the subjects did not know their lateral flow device results, minimizing potential bias, but leaves open the possibility that performance on finger prick samples might differ from that in the laboratory setting that we used. This concern has been addressed for the SureScreen device, for which data have now been published showing very similar accuracy on finger-prick samples taken from individuals and serum samples analyzed in a laboratory (25).

Finally, some health care workers in the cohort received a single vaccine dose in late December 2020 and early January 2021 in the United Kingdom, in the final weeks of our follow-up period; it is possible that some individuals acquired protection through vaccination in the last weeks of follow-up, so the true protection associated with LFIA-positive tests may be higher than we observed.

Lateral flow devices have very different reported accuracy. This is likely to be explicable in part by device design and in part by selection of samples used for device evaluation (21, 30, 31). Of the four LFIAs included in our study, comparative testing on a large, well-characterized sample set showed specificity was highest in SureScreen (98.9%; 95% CI, 98.3% to 99.3%) and AbC-19 (97.9%; 95% CI, 97.2% to 98.4%) and lower in Biomerica (97.3%; 95% CI, 96.5% to 98.0%) and OrientGene (96.9%; 95% CI, 96.1% to 97.6%) (21). Conversely, sensitivity

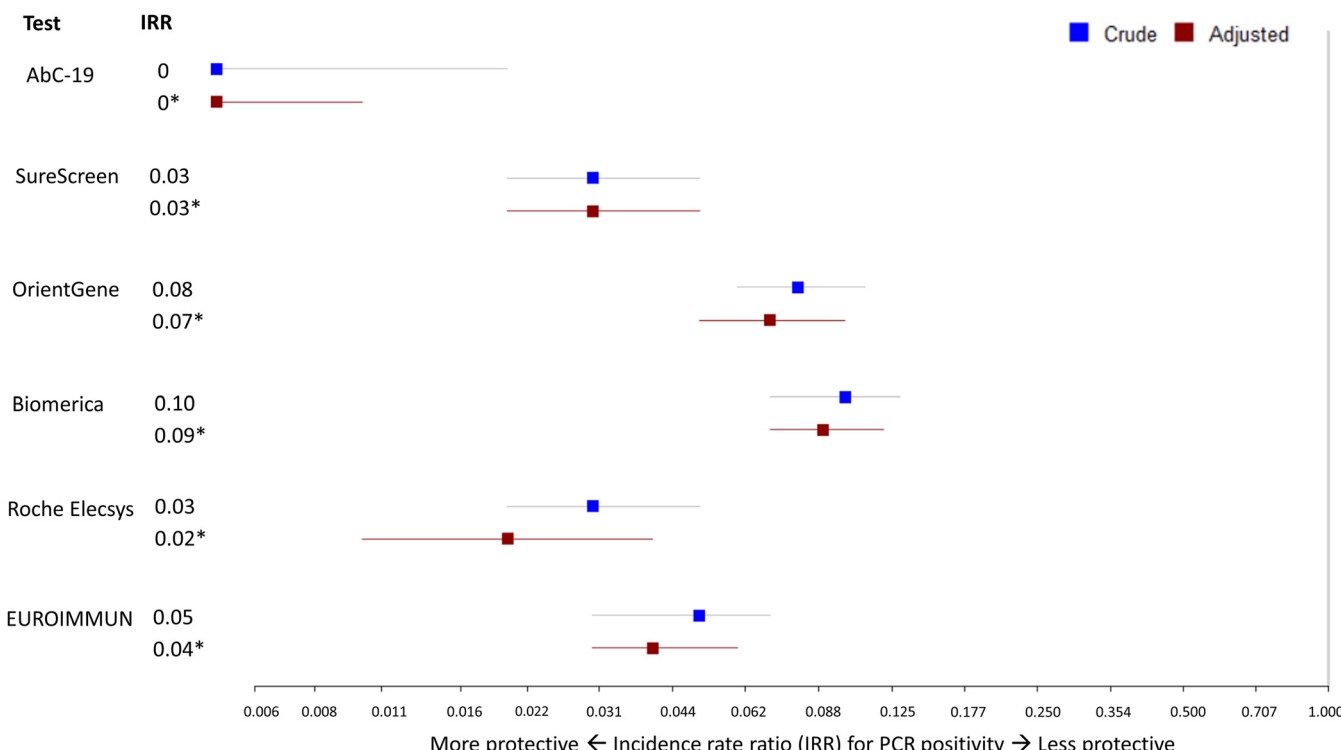

**FIG 3** Incidence rate ratios for SARS-CoV-2 PCR positivity by test result. Forest plot showing observed and adjusted (*) incidence rate ratios (IRRs, aIRRs) (95% confidence intervals) for PCR positivity in those who tested antibody positive in each test compared to those who tested antibody negative. Adjustment included sex, ethnicity (white or nonwhite), age (continuous), high-risk occupation (medical or nursing staff), geographic region, and average weekly SARS-CoV-2 infection incidence rate in the NHS region of residence of the participant. Roche Elecsys results of an immunoassay signal greater than 1.0 were considered positive. Euroimmun results of an immunoassay index greater than 0.8 were considered positive.

was lower in AbC-19 (92.5%; 95% CI, 88.8% to 95.1%) and SureScreen (88.8%; 95% CI, 84.5% to 92.0%) and higher in Biomerica (94.4%; 95% CI, 91.0% to 96.6%) and OrientGene (94.0%; 95% CI, 90.5% to 96.3%), which could detect lower levels of SARS-CoV-2 antibodies, including levels at which disease risk is elevated (Fig. 4) (21). SureScreen was estimated to have the highest positive predictive value and OrientGene and Biomerica the highest negative predictive values when detecting antibody at the manufacturer's cutoff value and relative to an ELISA-based gold standard (21). Therefore, the differential sensitivities of these devices may explain the variation in disease risk associated with testing LFIA positive using various different lateral flow devices. This observation may inform device choice and design decisions when lateral flow devices are optimized: during development, specificity/sensitivity trade-offs operate. Devices that are less sensitive but more specific, and which perform well in useability tests (25), may be more useful for predicting disease protection.

Overall, these data suggest that the more specific LFIA devices used here may have a role in surveillance programs assessing population protection in unvaccinated individuals, informing the debate about the risk to populations and perhaps in individual risk assessment (4, 5). In the context of higher vaccination rates, LFIAs could also play a role in assessing protection after vaccination and prioritizing the delivery of booster vaccines to groups with lower antibody levels. An ongoing program of field study would be required to show whether the LFIA-associated protection seen in this study extends to self-read and health care worker-read tests, to currently prevalent SARS-CoV-2 strains which may differ from the strains circulating during this study (32, 33), and (if surveillance of vaccinated individuals were contemplated) to vaccinated individuals.

## MATERIALS AND METHODS

**Study participants.** The study population consisted of key workers recruited to the Evaluating Detection of SARS-CoV-2 AntiBodies at HOME (EDSAB-HOME) prospective cohort study (ISRCTN56609224) (34). Full descriptions of the participant characteristics, sample size considerations, and recruitment methods have

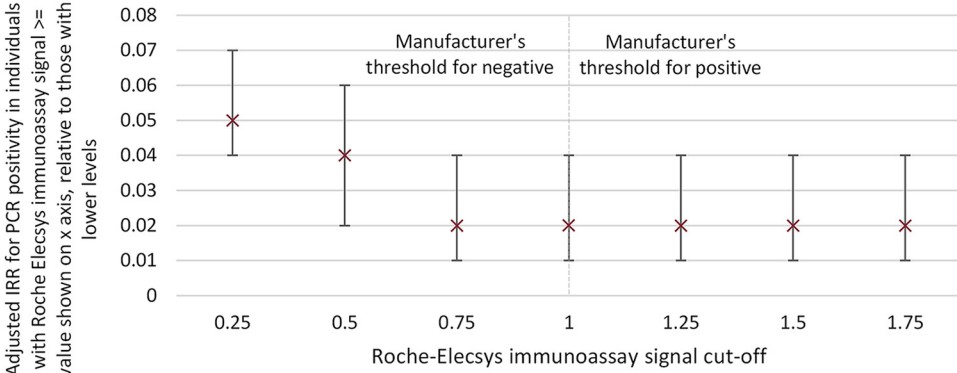

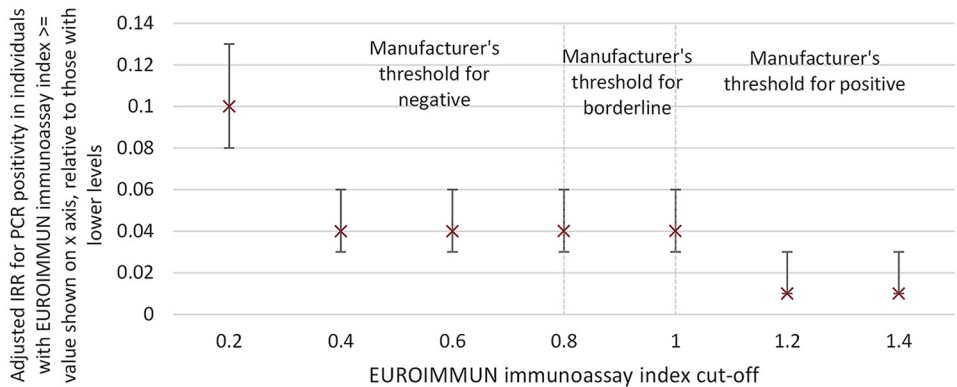

**FIG 4** Roche Elecsys and Euroimmun results and PCR positivity. Adjusted incidence rate ratios (IRRs) for PCR positivity when Roche Elecsys (top) and Euroimmun (bottom) assays, measuring anti-nucleoprotein and anti-spike S1 antibody reactivity, respectively, are dichotomized at a series of different thresholds. For example, the incidence rate ratio shown on the x axis at 1 refers to the incidence rate ratio in individuals with a baseline antibody level of 1 or higher, relative to those below. Adjustment included sex, ethnicity (white or nonwhite), age (continuous), high-risk occupation (medical or nursing staff), geographic region, and average weekly incidence rate in the NHS region of residence of the participant. Error bars show 95% confidence intervals.

been described previously (20–23, 26). Participants were recruited through their workplace into three streams: (A) police and fire & rescue service keyworkers; (B) health care keyworkers, both recruited regardless of previous SARS-CoV-2 infection status; and (C) health care workers purposely recruited due to a history of previous RT-PCR positivity. The cohort contained 2,847 participants as follows: 1,147 from police and fire (stream A), 1,546 health care workers (HCW) (stream B); and 154 from the health care worker previously COVID-19-positive test group (HCW-PP) (stream C). Participants were recruited from two non-health care worker sites (one police station in Hutton, Lancashire, and one fire and rescue center in Euxton, Lancashire) and six NHS acute hospitals (located in Milton Keynes, Gloucester, Cheltenham, York, Scarborough, and Rotherham), from 1 June to 26 June 2020 (35). We removed 21 individuals with incomplete laboratory test results, resulting in a total of 2,826 participants included in analysis. Recruitment took place in England in June 2020, at which point blood samples from participants were collected.

**Study endpoints.** The cohort was monitored for the development of microbiologically confirmed SARS-CoV-2 infection for the 200 days up to 24 January 2021 by linkage to a national database of SARS-CoV-2 results. Follow-up was stopped at this point due to the introduction of SARS-CoV-2 vaccination in the United Kingdom from December 2020 onwards (35).

The endpoint was testing positive for SARS-CoV-2 using nasal/throat PCR tests. In the period after recruitment, symptom-driven nasal/throat RT-PCR testing was available through state and employer routes for those with cough, fever, or disordered taste/smell; all such tests, irrespective of result, were recorded in a national database. Asymptomatic and lateral flow-based testing was available at the time, with positive lateral flow tests triggering PCR confirmation. Positive PCR tests triggered phone contact from NHS Test and Trace, a contact tracing service, who collected details of illness.

We obtained details about symptoms and circumstances associated with the positive test from (i) an optional weekly symptom questionnaire sent to volunteers, (ii) a symptom questionnaire sent retrospectively to all individuals with positive tests, and (iii) NHS Test and Trace data.

**Lateral flow immunoassays.** Four lateral flow immunoassay (LFIA) devices were described in this study as follows: the UK Rapid Test Consortium's AbC-19 rapid test (AbC-19), OrientGene COVID IgG/IgM rapid test cassette (OrientGene); SureScreen COVID-19 rapid test cassette (SureScreen); and Biomerica COVID-19 IgG/IgM rapid test (Biomerica) (20, 21). These devices had been selected by the UK Department of Health and Social Care's New Tests Advisory Group on the basis of test and performance data available, and our previous research has described their sensitivity and specificity (21). Blood samples collected at recruitment from participants were analyzed in the laboratory using four different lateral flow devices. Each device provides a qualitative positive or negative result. The AbC-19, OrientGene, and SureScreen devices detect anti-spike protein antibodies, while the Biomerica device detects anti-nucleoprotein antibodies. The OrientGene, SureScreen, and Biomerica devices contain separate bands to detect IgG and IgM antibodies, while AbC-19 detects IgG antibodies only. In our analyses, we classified test results from OrientGene, SureScreen, or Biomerica as positive if the IgG band was positive, disregarding the IgM bands. The laboratory protocol for conducting the lateral flow and laboratory-based immunoassays has been described elsewhere (21).

**Laboratory assays.** Blood samples from participants were also analyzed with two commercial laboratory immunoassays: Roche Elecsys anti-SARS-CoV-2 (nucleocapsid [N]) and Euroimmun anti-SARS-CoV-2 ELISA (IgG) assays (spike [S] protein S1 domain) (36, 37). Roche Elecsys results were dichotomized at the manufacturer recommended threshold of 1.0. For Euroimmun, an immunoassay index lower than 0.8 is defined by the manufacturer as negative, an index between 0.8 and 1.0 is considered borderline, and an index greater than 1.0 is positive. In order to dichotomize the results, in this study an immunoassay index greater than 0.8 for Euroimmun was defined as positive.

**Blinding.** The individuals who conducted the laboratory or lateral flow immunoassays could not access information about the samples or results on any other assay. Participants were informed of Euroimmun serological results approximately 1 month after visiting the clinic, with a warning that this was not indicative of protection from disease. Participants were not informed of their Roche or LFIA results.

**Statistical analyses.** Statistical analyses were conducted in R version 1.3.1056. Participant follow-up was divided into three periods based on national data describing the different waves of infection in the population at that time (Fig. 1). For each antibody test, we describe the number of participants who tested positive for SARS-CoV-2 and observed rate per 100 person years in each of these three periods, stratified by baseline test result. Poisson regression modeling with Firth penalization was used to estimate incidence rates and incidence rate ratios (IRRs), describing the rate of SARS-CoV-2 positivity in individuals who tested antibody positive compared to those who tested antibody negative for each test and confidence intervals. Adjusted IRRs were computed, by modeling in addition baseline risk factors for SARS-CoV-2 acquisition, specifically sex, ethnicity (white or nonwhite), age (continuous), high-risk occupation (medical or nursing staff), geographic region, and time-updated regional SARS-CoV-2 infection rates (average weekly incidence in the NHS region of residence of the participant). Kaplan-Meier survival curves were also produced to describe the development of SARS-CoV-2 positivity over time in those who were antibody positive or antibody negative according to each test.

In additional exploratory analyses, the relationship between the thresholds used to define antibody positivity for the Roche Elecsys and Euroimmun laboratory-based immunoassays and the adjusted IRRs for PCR positivity was explored graphically.

**Ethics.** The EDSAB-HOME study was approved by the NHS Research Ethics Committee (Health Research Authority, IRAS 284980) on 2 June 2020 and PHE Research Ethics and Governance Group (REGG, NR0198) on 21 May 2020. All participants gave written informed consent.

**Data availability.** An anonymized data set comprising lateral flow device results, laboratory immunoassay results, and a dichotomous outcome (SARS-CoV-2 PCR result) is available as Data Set S1 in the supplemental material. Researchers with requests for more granular data, including individual risk factors and dates of events, should contact the authors, as a risk assessment regarding individual identifiability will need to be made.

## SUPPLEMENTAL MATERIAL

Supplemental material is available online only.
**SUPPLEMENTAL FILE 1**, XLSX file, 0.1 MB.

## ACKNOWLEDGMENTS

The study was commissioned by the UK Government's Department of Health and Social Care and funded and implemented by Public Health England, supported by the NIHR Clinical Research Network (CRN) Portfolio.

The Department of Health and Social Care had no role in the study design, data collection, analysis, interpretation of results, writing of the manuscript, or the decision to publish.

This study was funded in part by the National Institute for Health and Care Research (NIHR) Health Protection Research Unit in Genomics and Enabling Data (NIHR200892), a partnership between the UK Health Security Agency, the University of Warwick, the University of Oxford and the University of Cambridge. L. Findlater, H. E. Jones, A. Trickey, and I. Oliver acknowledge support from the NIHR Health Protection Research Unit in Behavioural Science and Evaluation at University of Bristol. A. Trickey is supported by a

Wellcome Trust Sir Henry Wellcome Fellowship. S. Taylor-Phillips is supported by an NIHR Career Development Fellowship (CDF-2016-09-018).

The views expressed are those of the author(s) and not necessarily those of the NHS, NIHR, or the Department of Health and Social Care.

Conceptualization, H. E. Jones, S. Taylor-Phillips, R. Mulchandani, I. Oliver, and D. Wyllie; Data Curation, A. Trindall, D. Wyllie, and R. Mulchandani; Formal Analysis, L. Findlater, A. Trickey, and D. Wyllie; Funding, I. Oliver; Investigation, A. Trickey, R. Mulchandani, and D. Wyllie; Methodology, A. Trickey, H. E. Jones, S. Taylor-Phillips, and D. Wyllie; Resources, Philippa Moore, John Boyes, Anil Hormis, Neil Todd, and Ian Reckless; Writing – first draft, L. Findlater, D. Wyllie. EDSAB-HOME clinical investigators (Philippa Moore, John Boyes, Anil Hormis, Neil Todd, and Ian Reckless) contributed to study design and resources (supporting access to volunteers). Reviewed and revised manuscript, all authors. Access to underlying data, R. Mulchandani, D. Wyllie, and L. Findlater.

EDSAB-HOME investigators: Philippa Moore and John Boyes (Gloucestershire Hospitals NHS Foundation Trust), Anil Hormis (The Rotherham NHS Foundation Trust), Neil Todd (York Teaching Hospital NHS Foundation Trust), and Ian Reckless (Milton Keynes University Hospital NHS Foundation Trust).

We declare that we have no conflicts of interest.

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
