## [Reviewer comments · Microbiology Spectrum]

Microbiology Spectrum

Association of results of four lateral flow antibody tests with subsequent SARS-CoV-2 infection

Lucy Findlater, Adam Trickey, Hayley Jones, Amy Trindall, Sian Taylor-Phillips, Ranya Mulchandani, EDSAB-HOME Investigators, Isabel Oliver, and David Wyllie

Corresponding Author(s): David Wyllie, Public Health England

Review Timeline:

Submission Date:	July 11, 2022
Editorial Decision:	August 8, 2022
Revision Received:	August 22, 2022
Accepted:	September 6, 2022

Editor: Heba Mostafa

Reviewer(s): The reviewers have opted to remain anonymous.

Transaction Report:

DOI: <https://doi.org/10.1128/spectrum.02468-22>

August 8, 2022

Dr. David H Wyllie
Public Health England
Cambridge
United Kingdom

Re: Spectrum02468-22 (Association of results of four lateral flow antibody tests with subsequent SARS-CoV-2 infection)

Dear Dr. David H Wyllie:

Link Not Available

Sincerely,

Heba Mostafa

Journals Department
Reviewer comments:

Reviewer #1 (Comments for the Author):

Overall this paper sets out to address an important, unresolved, question and reads very clearly.

I only have a few minor suggestions for modification which might help strengthen the paper

Rationale: the potential for an LFIA based approach is probably stronger to inform vaccination policy (e.g. by identifying groups with lower response) than individual management. The intro/discussion could be strengthened on this as well as relevance of these findings to vaccination induce immune responses

It would be helpful to provide the performance characteristics of the tests within the paper to help follow the

discussion/interpretation

The KM graphs are helpful. What would be the authors explanation for these plateauing? Could this reflect access to vaccination? Worth discussion?

Could KMs be generated for the lab assays (at recommended threshold, and potentially alternative thresholds)

Minor:

Line 231 should read "NOT"?

Line 242, missing?

Refs - need some formatting e.g. refs 1,7

The authors may be interested in this recent paper:10.1093/cid/ciac629

Reviewer #2 (Comments for the Author):

The authors evaluated a large (N=2826) cohort of plasma taken from individuals recruited between June 2020 and January 2021 of the SARS-CoV-2 pandemic to determine if the use of rapid immunoassays could provide data indicating immune protection (or diseases susceptibility) from SARS-CoV-2. Their results indicate that positive IgG results were associated with lower rates of subsequent infection. The sample size, methodology and statistical methods used in this study are appropriate.

Major Comment: Given that this study was conducted in the first year of the pandemic before vaccines were widely available, and also prior to the emergence of the Delta, Omicron and BA.5 variants, the authors need to provide additional commentary regarding the applicability of this data in late 2022. As the authors mention, ~65% of the world's population have received at least one vaccine dose, so this data may no longer apply to that population, and while it's true that vaccination rates are substantially lower in low income countries, those individuals are now going to be exposed to BA.5, or possibly another emerging variant. Do the authors have data indicating that the LFIA's employed in this study perform similarly well for BA.5? What scenarios do the authors envision for applying this methodology to the broader population now that vaccination rates are higher overall? Additionally, what was the rationale behind the LFIA's chosen for this study, was it a matter of convenience? Did the authors perform a comparison study with other LFIA's and chose the most sensitive and specific assays? What is the performance data , ie, sensitivity, specificity, ppv, npv of these tests?

Staff Comments:

Preparing Revision Guidelines

Please return the manuscript within 60 days; if you cannot complete the modification within this time period, please contact me. If you do not wish to modify the manuscript and prefer to submit it to another journal, please notify me of your decision immediately so that the manuscript may be formally withdrawn from consideration by Microbiology Spectrum.

Response to Reviewers

Reviewer 1

Overall, this paper sets out to address an important, unresolved, question and reads very clearly. I only have a few minor suggestions for modification which might help strengthen the paper

- Thank you very much for your review. We have made the following changes below.

Rationale: the potential for an LFIA based approach is probably stronger to inform vaccination policy (e.g., by identifying groups with lower response) than individual management. The intro/discussion could be strengthened on this as well as relevance of these findings to vaccination induce immune responses

- We have added a sentence to the first paragraph of the introduction: "In vaccinated populations, understanding individual risk could also help to monitor immune responses to vaccination and inform prioritisation of booster delivery".
- We have also added a sentence to the last paragraph of the discussion: "In the context of higher vaccination rates, LFIA's could also play a role in assessing protection after vaccination and prioritising the delivery of booster vaccines to groups with lower antibody levels".

It would be helpful to provide the performance characteristics of the tests within the paper to help follow the discussion/interpretation

- We have added a section in the discussion (line 280) which describes the sensitivity and specificity of each of the LFIA's, and states which LFIA's had the highest NPV and PPV: "Of the four LFIA's included in our study, comparative testing on a large, well-characterised sample set showed specificity was highest in SureScreen (98.9%, 95% CI 98.3% to 99.3%) and AbC-19 (97.9%, 95% CI 97.2% to 98.4%) and lower in Biomerica (97.3%, 95% CI 96.5% to 98.0%) and OrientGene (96.9%, 95% CI 96.1% to 97.6%) (21). Conversely, sensitivity was lower in AbC-19 (92.5%, 95% CI 88.8% to 95.1%) and SureScreen (88.8%, 95% CI 84.5% to 92.0%), and higher in Biomerica (94.4%, 95% CI 91.0% to 96.6%) and OrientGene (94.0%, 95% CI 90.5% to 96.3%), which could detect lower levels of SARS-CoV-2 antibodies, including levels at which disease risk is elevated (Figure 4)(21). SureScreen was estimated to have the highest positive predictive value, and OrientGene and Biomerica the highest negative predictive values, when detecting antibody at the manufacturer's cut-off value and relative to an ELISA-based gold standard (21)."

The KM graphs are helpful. What would be the authors explanation for these plateauing? Could this reflect access to vaccination? Worth discussion?

- Thank you for your comment. With regard to the plateauing observed in the KM graphs, this could possibly reflect decline in incidence of SARS-CoV-2, population protection being acquired, or increased vaccination. However, we have declined to comment in the paper.

Could KMs be generated for the lab assays (at recommended threshold, and potentially alternative thresholds)

- We have generated KMs for the Roche and EuroImmun lab assays at the recommended threshold and added them to Figure 2, alongside the KMS for the LFIA's.

Line 231 should read "NOT"?

- We have corrected this typo and it now reads “than individuals who did not have detectable antibodies”.

Line 242, missing?

- We have corrected this typo and it now reads “This concern has been addressed for the SureScreen device, for which data has now been published showing very similar accuracy on finger-prick samples taken from individuals and serum samples analysed in a laboratory.”

Refs - need some formatting e.g. refs 1,7

- We have improved the formatting of the references such that the organisation is correctly displayed.

The authors may be interested in this recent paper:[10.1093/cid/ciac629](https://doi.org/10.1093/cid/ciac629)

- Thank you for highlighting this very relevant paper. We have now cited it in the introduction and the discussion.

Reviewer 2

The authors evaluated a large (N=2826) cohort of plasma taken from individuals recruited between June 2020 and January 2021 of the SARS-CoV-2 pandemic to determine if the use of rapid immunoassays could provide data indicating immune protection (or diseases susceptibility) from SARS-CoV-2. Their results indicate that positive IgG results were associated with lower rates of subsequent infection. The sample size, methodology and statistical methods used in this study are appropriate.

- Thank you very much for your review. We have made the following changes below.

Major Comment: Given that this study was conducted in the first year of the pandemic before vaccines were widely available, and also prior to the emergence of the Delta, Omicron and BA.5 variants, the authors need to provide additional commentary regarding the applicability of this data in late 2022. As the authors mention, ~65% of the world's population have received at least one vaccine dose, so this data may no longer apply to that population, and while it's true that vaccination rates are substantially lower in low income countries, those individuals are now going to be exposed to BA.5, or possibly another emerging variant. Do the authors have data indicating that the LFIAs employed in this study perform similarly well for BA.5?

- We agree that this is an important point to discuss and have added more detail in the second paragraph of the discussion to highlight this as an important caveat: “Firstly, it applies to a historical cohort of unvaccinated individuals and prior to the emergence of variants such as Delta, Omicron, and BA.5 (33). The performance of LFIAs might be different in the context of immunity gained from vaccination (in addition to or instead of natural infection) and to estimate protection against different circulating SARS-CoV-2 variants with differential antibody neutralisation; this is an important area for future research (8).”
- We have also emphasised the need for further research to explore if LFIAs perform similarly well for other variants in the final paragraph of the discussion: “An ongoing programme of field studies would be required to show whether the LFIAs-associated protection seen in this study extends to self-read and healthcare worker-read tests; to currently prevalent SARS-CoV-2 strains

which may differ from the strains circulating during this study (36, 37); and (if surveillance of vaccinated individuals were contemplated) to vaccinated individuals.”

What scenarios do the authors envision for applying this methodology to the broader population now that vaccination rates are higher overall?

- We have added a sentence to the first paragraph of the introduction highlighting that LFIA's could still play a role in the context of higher vaccination rates: “In vaccinated populations, understanding individual risk could also help to monitor immune responses to vaccination and inform prioritisation of booster delivery (5-8)”.
- We have also emphasised this in the final paragraph of the discussion: “In the context of higher vaccination rates, LFIA's could also play a role in assessing protection after vaccination and prioritising the delivery of booster vaccines to groups with lower antibody levels.”

Additionally, what was the rationale behind the LFIA's chosen for this study, was it a matter of convenience? Did the authors perform a comparison study with other LFIA's and chose the most sensitive and specific assays?

- We have added the rationale behind choosing these LFIA's in the ‘Lateral flow immunoassays’ paragraph of the Methods section: “These devices had been selected by the UK Department of Health and Social Care’s New Tests Advisory Group on the basis of test and performance data available, and our previous research has described their sensitivity and specificity (21)”

What is the performance data, ie, sensitivity, specificity, ppv, npv of these tests?

- We have added a section in the Discussion describing the performance data and citing our previous paper which contains more detailed information on test performance: “Of the four LFIA's included in our study, comparative testing on a large, well-characterised sample set showed specificity was highest in SureScreen (98.9%, 95% CI 98.3% to 99.3%) and AbC-19 (97.9%, 95% CI 97.2% to 98.4%) and lower in Biomerica (97.3%, 95% CI 96.5% to 98.0%) and OrientGene (96.9%, 95% CI 96.1% to 97.6%) (21). Conversely, sensitivity was lower in AbC-19 (92.5%, 95% CI 88.8% to 95.1%) and SureScreen (88.8%, 95% CI 84.5% to 92.0%), and higher in Biomerica (94.4%, 95% CI 91.0% to 96.6%) and OrientGene (94.0%, 95% CI 90.5% to 96.3%), which could detect lower levels of SARS-CoV-2 antibodies, including levels at which disease risk is elevated (Figure 4)(21). SureScreen was estimated to have the highest positive predictive value, and OrientGene and Biomerica the highest negative predictive values, when detecting antibody at the manufacturer's cut-off value and relative to an ELISA-based gold standard (21).”

September 6, 2022

Dr. David H Wyllie
Public Health England
Cambridge
United Kingdom

Re: Spectrum02468-22R1 (Association of results of four lateral flow antibody tests with subsequent SARS-CoV-2 infection)

Dear Dr. David H Wyllie:

Your manuscript has been accepted, and I am forwarding it to the ASM Journals Department for publication. You will be notified when your proofs are ready to be viewed.

Sincerely,

Heba Mostafa
Editor, Microbiology Spectrum

Journals Department
Supplemental Dataset: Accept